

# Evaluation of aerosol number concentrations from CALIPSO with ATom airborne in-situ measurements

Goutam Choudhury[1], Albert Ansmann[2], and Matthias Tesche[1]

[1]Leipzig Institute for Meteorology, Leipzig University, Leipzig, Germany
[2]Leibniz Institute for Tropospheric Research, Leipzig, Germany

**Correspondence:** Goutam Choudhury (goutam.choudhury@uni-leipzig.de)

**Abstract.** The present study aims to evaluate the available aerosol number concentration (ANC) retrieval algorithms for space-borne lidar CALIOP (Cloud-Aerosol Lidar with Orthogonal Polarization) aboard CALIPSO (Cloud-Aerosol Lidar and Infrared Pathfinder Satellite Observations) satellite with the airborne in-situ measurements from ATom (Atmospheric Tomography Mission) campaign. We used HYSPLIT (Hybrid Single-Particle Lagrangian Integrated Trajectory model) to match both the measurements in space and identified 53 cases that were suitable for comparison. Since the ATom data include dry aerosol extinction coefficient, we used kappa parameterization to adjust the ambient measurements from CALIOP to dry conditions. As both the datasets have a different vertical resolution, we re-grid them to uniform height bins of 240 m from the surface to a height of 5 km. On comparing the dry extinction coefficients, we found a reasonable agreement between the CALIOP and ATom measurements with Spearman's correlation coefficient of 0.715. Disagreement was found mostly for retrievals above 3 km altitude. Thus, to compare the ANC which may vary orders of magnitude in space and time, we further limit the datasets and only select those height bins for which the CALIOP derived dry extinction coefficient is within $\pm 50$ % of the ATom measurements. This additional filter further increases the probability of comparing the same air parcel. The altitude bins which qualify the extinction coefficient constraint are used to estimate ANC with dry radius >50 nm ($n_{50,\mathrm{dry}}$) and >250 nm ($n_{250,\mathrm{dry}}$). The POLIPHON (Polarization Lidar Photometer Networking) and OMCAM (Optical Modelling of CALIPSO Aerosol Micro-physics) algorithms were used to estimate the $n_{50,\mathrm{dry}}$ and $n_{250,\mathrm{dry}}$. The POLIPHON estimates of $n_{50,\mathrm{dry}}$ and $n_{250,\mathrm{dry}}$ were found to be in good agreement with the in-situ measurements with a correlation coefficient of 0.829 and 0.47, respectively. The OMCAM estimates of $n_{50,\mathrm{dry}}$ and $n_{250,\mathrm{dry}}$ were also in reasonable agreement with the in-situ measurements with a correlation coefficient of 0.823 and 0.463, respectively. However, we found that the OMCAM estimated $n_{50,\mathrm{dry}}$ were about an order less than the in-situ measurements for marine dominated cases. We propose a modification to the OMCAM algorithm by using an AERONET-based marine model. With the updated OMCAM algorithm, the $n_{50,\mathrm{dry}}$ agree well with the ATom measurements. Such concurrence between the satellite-derived ANC and the independent ATom in-situ measurements emboldens the use of CALIOP in studying the aerosol-cloud interactions.





# 1 Introduction

Aerosol particles are needed to form clouds under the majority of atmospheric conditions. They can act as cloud condensation
nuclei (CCN) initiating liquid droplet nucleation in warm clouds and as ice nucleating particles (INPs) initiating heterogeneous
ice nucleation in mixed-phase and cold clouds. Changes in the concentration of such particles influence the cloud extent,
development, lifetime, and microphysical and radiative properties (Fan et al., 2016; Seinfeld et al., 2016; Choudhury et al.,
2019). Inadequate understanding of such complex aerosol-cloud interactions (ACIs) and the corresponding rapid adjustments
in radiative forcing are key reasons behind the uncertainty in our future climate projections (IPCC, 2021).

The CCN and INP concentrations are the fundamental aerosol parameters needed to study the ACIs. A comprehensive rep-
resentation of the same in the weather and climate models is necessary to obtain a realistic simulation of the impact of aerosols
on cloud microphysics and the corresponding adjustments. By comparing the simulations from a total of 16 general circulation
models and global chemistry transport models with the in-situ measurements from 9 ground-based stations, Fanourgakis et
al. (2019) found that the models underestimate the aerosol number and CCN concentrations. Similar underestimation is also
reported by Genz et al. (2020). Compared to CCN which may vary from anywhere between $10^2$ to $10^5$ cm$^{-3}$, INPs are sparse
in nature with about one in a million particles capable of forming ice crystals in the atmosphere (Nenes et al., 2014). By
combining the GLOMAP-mode global aerosol microphysics model (Mann et al., 2010) and field experiments of k-feldspar
and marine organic aerosols, Vergara-Temprado et al. (2017) compared the INP concentrations with in-situ measurements at
marine locations and found the annual mean modelled INP values to be 1.5 orders of magnitude larger than the observations.
Also, depending on the INP parametrization and temperature of measurement, the modelled INP concentrations can be as much
as 4-6 orders of magnitude larger than the observations (Vergara-Temprado et al., 2017). Thus, a better global measurement
of cloud-relevant aerosol microphysical properties is needed for constraining our weather and climate models. While the sur-
face in-situ measurements of such parameters are carried out continuously with a high temporal resolution, they are limited
to certain point locations. One way to overcome this limitation is to switch to satellite observations, which provide global,
continuous, and long-term monitoring of the atmosphere.

Satellite retrievals used in ACI studies include aerosol optical parameters like the aerosol backscatter coefficient, aerosol
extinction coefficient, and aerosol optical depth (column integrated aerosol extinction coefficient). Compare to the column
integrated products obtained from passive sensors, active sensors like lidars provide height resolved optical parameters which
are necessary for studying vertically collocated aerosols and clouds (Costantino and Bréon, 2013). Cloud-Aerosol Lidar with
Orthogonal Polarization (CALIOP) is a spaceborne lidar aboard the Cloud-Aerosol Lidar and Infra-Red Pathfinder Satellite
Observations (CALIPSO) satellite, which provides profiles of aerosol optical parameters like the backscatter coefficient, ex-
tinction coefficient, and particle depolarization ratio. Recent studies have shown that these optical parameters can be used
to derive cloud-relevant aerosol microphysical parameters. Mamouri and Ansmann (2016) present the first CCN and INP re-
trieval algorithm for measurements with ground-based lidars. The algorithm includes two main steps: (1) the conversion of
the lidar-derived extinction coefficient to aerosol number concentration (ANC) with dry radii >50 nm ($n_{50,\mathrm{dry}}$) and >250 nm
($n_{250,\mathrm{dry}}$) and (2) the subsequent use of the ANC estimates to compute CCN (at different supersaturations) and INPs (at differ-





ent temperatures) concentrations based on aerosol-type specific parameterizations. The parameterizations for estimating CCN concentrations for different aerosol types are given in (Mamouri and Ansmann, 2016) and those for INP are available, e.g., in DeMott et al. (2010, 2015) and Ullrich et al. (2017). Though the methodology for retrieving CCN and INP concentrations

was developed for ground-based lidars, it has also been applied to measurements with the spaceborne lidar CALIOP (Marinou et al., 2019; Georgoulias et al., 2020). This highlight the potential of CALIOP for estimating global 3D CCN and INP concentrations for climatological data sets. More recently, Choudhury and Tesche (2022) presented a CCN/INP retrieval algorithm developed specifically for CALIOP measurements. It uses the aerosol-type specific normalized size distributions from the CALIPSO aerosol model (Omar et al., 2009) and scales them as per the extinction coefficient measured by CALIOP. The

final size distribution is integrated to get the ANC required in the CCN and INP parameterizations. Of key importance is the accurate retrieval of ANC from satellites – the primary component of CCN and INP parameterizations. However, a thorough validation of the same is missing except for selected case studies (Marinou et al., 2019; Georgoulias et al., 2020).

The Atmospheric Tomography Mission (ATom, Wofsy et al. (2018)) comprises a series of continuous flight measurements over different parts of the Pacific and Atlantic oceans from 2016 to 2018 measuring aerosol properties including the ANC. This

dataset provides a unique opportunity to validate the available ANC retrieval algorithms for the spaceborne lidar CALIOP. In this study, we validate the ANC retrieval algorithms presently available for CALIOP measurements with the airborne in-situ measurements from the ATom campaigns. Moreover, we suggest a revision to the ANC retrieval algorithm given by Choudhury and Tesche (2022). This manuscript is organized as follows. The description of the datasets, ANC retrieval algorithms for CALIOP, and the comparison methodology are given in Section 2. The results are presented in Section 3 and discussed in

Section 4. The main findings are summarized in Section 5.

## 2 Data, retrievals, and methods

### 2.1 ATom

The ATom comprised of four series of flights by the NASA DC-8 research aircraft over the Pacific and Atlantic oceans covering latitudes between 82°N and 86°S. The flight patterns included regular descents and ascents between altitudes of 200 m and 12

80 km. A total of four ATom campaigns were conducted between August and September 2016 (ATom1), January and February 2017 (ATom2), September and October 2017 (ATom3), and April and May 2018 (ATom4). The instruments employed for measuring the dry aerosol particle size distribution between a radius of 1.35 nm and 2.4 µm are a Laser Aerosol Spectrometer (LAS), a Nucleation-mode Aerosol Size Spectrometer (NMASS), and an Ultra-high Sensitivity Aerosol Size Spectrometer (UHSAS). The operating principles of these instruments and their inferred data products are described comprehensively in

Brock et al. (2019). In the present study, we use version 1.5 of ATom: Merged Atmospheric Chemistry, Trace Gases, and Aerosols dataset (Wofsy et al., 2018) with a very high temporal resolution of 10 s. The parameters used in our comparison study are given in Table 1. To compute $n_{50,\mathrm{dry}}$, we add the LAS measured number concentration in the accumulation mode (0.05<R<0.4425 µm) and coarse mode (0.4425≤R< approx. 2 µm). During ATom2, a leak was found in the sheath flow of the LAS, leading to lower detection efficiency. Simultaneous measurements from other instruments were used to correct the LAS





measurements (Brock et al., 2019). The extinction coefficient is calculated from the dry size distributions by using Mie theory (Bohren and Huffman, 2008) assuming that particles are composed of homogenous non-absorbing spheres of ammonium sulfate with a refractive index of 1.52. Note that this extinction coefficient is reported for particles with dry radius R≤2.4 µm. Coarse particles with a dry radius R>2.4 µm may contribute significantly to the extinction coefficient within the marine boundary layer and dust-dominated air masses (Brock et al., 2019). The extinction coefficient in such scenarios is likely to be 95 underestimated. All ATom parameters are given at standard temperature and pressure.

## 2.2 CALIOP

CALIOP is a dual-wavelength, three-channel polarization-sensitive lidar aboard the polar-orbiting CALIPSO (Winker et al., 2009) satellite that was launched on April 28, 2006, as a part of the A-Train constellation. CALIOP provides global height-resolved coverage of the occurrence and properties of aerosol and cloud layers. For inferring aerosol backscatter and extinction 100 coefficients, the CALIPSO retrieval requires a priori information on the prevailing aerosol type. The aerosol types defined in the CALIPSO v4 retrieval algorithm include *marine, desert dust, polluted continental/smoke, clean continental, elevated smoke, polluted dust*, and *dusty marine*. A respective aerosol type is selected by considering the estimated 532 nm particle depolarization ratio, the 532 nm integrated attenuated backscatter coefficient, the aerosol layer height (top and bottom), and the underlying surface type (Kim et al., 2018). For each detected aerosol type, the retrieval uses a pre-set type-specific lidar 105 ratio that has been estimated from a combination of long-term Aerosol Robotic Network (AERONET, Holben et al. (1998)) measurements and field campaigns with a subsequent adjustment based on independent measurements with ground-based lidars (Omar et al., 2005, 2009; Kim et al., 2018). The thus obtained profiles of the backscatter and extinction coefficient of aerosols and clouds are provided in the corresponding level 2 profile products. In the present study, we use the CALIPSO level 2 v4.20 aerosol profile product (CALIPSO, 2018). The parameters used to derive aerosol number concentrations from 110 CALIPSO measurements are the 532 nm aerosol extinction coefficient, the 532 nm aerosol backscatter coefficient, the 532 nm aerosol depolarization ratio, and the aerosol subtype mask. Quality control flags are used to select the most reliable data. To account for the hygroscopicity of aerosols, we use relative humidity (RH) profiles attained from the Global Modelling and Assimilation Office Data Assimilation System (Molod et al., 2015), included in the CALIPSO profile product. All parameters have a uniform horizontal resolution of 5 km and a vertical resolution of 60 m for tropospheric aerosols.

## 2.3 Aerosol number concentration from CALIOP

In this section, we discuss the two algorithms for estimating ANC from CALIOP measurements used in this study. In the analysis, we select only high-quality CALIOP data that fulfil the criteria given in Tackett et al. (2018) by utilizing the (i) cloud aerosol discrimination score (≤-20), the (ii) extinction quality check flag (=0, 1, 16, and 18), and the (iii) extinction uncertainty value (≠99.9). Also, for retrievals corresponding to the mixed aerosol types *polluted dust* and *dusty marine*, we 120 first separate the dust and non-dust contributions by using the particle depolarization ratio to separate the contributions of dust and non-dust aerosols to the particle backscatter coefficient (Tesche et al., 2009). The backscatter coefficient is then multiplied by the lidar ratio to yield dust and non-dust extinction coefficients. We use the updated lidar ratios from Kim et al. (2018). The





dust separation technique is also incorporated in many studies concerning the lidar-based retrieval of aerosol microphysical properties (Mamouri and Ansmann, 2015, 2016; Georgoulias et al., 2020; Choudhury and Tesche, 2022). The information on
aerosol-type specific extinction coefficient, aerosol type, and relative humidity are used to compute the ANC.

### 2.3.1 POLIPHON

The Polarization Lidar Photometer Networking (POLIPHON, Mamouri and Ansmann (2015, 2016)) method combines lidar-derived type-specific aerosol optical properties with concurrent long-term AERONET measurements of aerosol optical depth (AOD) and retrieved column size distributions (Dubovik et al., 2000, 2006) to estimate the ANC. A regression analysis of
130 the AERONET-derived column extinction coefficients and number concentrations (integral of the aerosol size distribution) yields the conversion equation to derive ANC from lidar-derived extinction coefficients. The regression analysis was based on AERONET observations at sites with pure marine or pure mineral dust conditions as well as observations in environments dominated by urban haze or wildfire smoke. The complex analysis resulted in aerosol type-specific conversion equations of the form

$$135 \quad n_{j,\mathrm{dry}} = C \cdot \alpha^x, \tag{1}$$

where $n_{j,\mathrm{dry}}$ is the aerosol number concentration with dry radius $>j$ nm, $\alpha$ is the extinction coefficient, $C$ is the conversion factor, and $x$ is the extinction exponent. In this study, we use the regression parameters for marine and continental aerosols given in Mamouri and Ansmann (2016). The one for desert dust is taken from Ansmann et al. (2019) and represent a global average. For smoke aerosols, we use the averaged value for aged smoke given in Ansmann et al. (2021) as most of the ATom
measurements were performed over oceans away from smoke sources. The values of the regression constants along with their sources are listed in Table 2. A typical RH of 80 % and 60 % were assumed while calculating the conversion factors for marine and continental (including smoke) aerosol types. Note that for dust aerosols, POLIPHON provides the ANC for dry radius R>100 nm ($n_{100,\mathrm{dry}}$) and we get $n_{50,\mathrm{dry}}$ from the ATom measurements.

### 2.3.2 OMCAM

The Optical Modelling of CALIPSO Aerosol Microphysics (OMCAM, Choudhury and Tesche (2022)) algorithm utilizes the normalized size distributions and refractive indices from the CALIPSO aerosol model (Omar et al., 2009) to derive those aerosol size distributions that lead to the best reproduction of the inferred aerosol extinction coefficient when used as input for light-scattering calculations with the MOPSMAP optical modelling package (Gasteiger and Wiegner, 2018). In the modelling of the extinction coefficient, we consider marine, continental, and smoke aerosols as spheres and apply the Mie scattering
theory. Mineral dust is considered to be spheroidal and is treated with a combination of the T-matrix method and the improved geometric optics method. The normalized size distribution from the CALIPSO aerosol model is scaled to reproduce the CALIOP-derived extinction coefficient (Choudhury and Tesche, 2022) as

$$\frac{\mathrm{d}V(r)}{\mathrm{d}\ln r} = \frac{\alpha}{\alpha_{\mathrm{n}}} \cdot \sum_{i=1}^{2} \frac{\nu_i}{\sqrt{2\pi}\ln\sigma_i} \exp\left(\frac{-(\ln r - \ln\mu_i)^2}{2\ln\sigma_i{}^2}\right). \tag{2}$$





where $\alpha_{\mathrm{n}}$ is the extinction coefficient estimated from the normalized size distribution and refractive index using MOPSMAP

and $\alpha$ is the CALIOP-derived extinction coefficient. $\sigma_i$, $\nu_i$, and $\mu_i$ are the standard deviation, volume fraction, and mean radius of the $i^{\mathrm{th}}$ mode of the normalized size distribution, respectively. Since the refractive index, $\sigma_i$, $\nu_i$, and $\mu_i$ of the aerosol size distributions are intensive parameters, i.e. independent of the extinction coefficient or aerosol concentration, the volume size distribution and hence the ANC can be expressed linearly in terms of the extinction coefficient as

$$n_{j,\mathrm{dry}} = C_{\mathrm{o}} \cdot \alpha \qquad (3)$$

where $C_{\mathrm{o}} = \frac{1}{\alpha_{\mathrm{n}}} \int_j^{r^{\max}} \mathrm{d}N$ is a conversion factor whose value depends on the aerosol type and the lower limit $j$ of integrating the particle size distribution. The values of $C_{\mathrm{o}}$ for different aerosol types are given in Table 3.

Choudhury and Tesche (2022) show a large discrepancy in their comparison of theoretically possible ANC for marine aerosols as estimated by POLIPHON and OMCAM. This can be attributed to the difference in the temporal extent and geographical location of the measurements, and different instruments employed in measuring of the marine size distributions

used in the two algorithms. The regression constants for marine aerosols used in POLIPHON are estimated from 7.5 years of AERONET measurements from 2007 to 2015 at Barbados (Mamouri and Ansmann, 2016). However, the marine model used in OMCAM (Omar et al., 2009; Choudhury and Tesche, 2022) was obtained from in-situ measurements of sea-salt size distributions during the SEAS experiment from 21 to 30 April 2000 (Masonis et al., 2003; Clarke et al., 2003). AERONET provides long-term continuous measurements of aerosol optical and microphysical properties at different locations around the

globe. Sayer et al. (2012) presented a maritime aerosol model for use in satellite retrievals based on the aerosol microphysical properties at 11 AERONET island stations. In this work, we utilize the microphysical properties recommended by Sayer et al. (2012) in the OMCAM algorithm and estimate the ANC separately to examine its potential in deriving the ANC from CALIOP measurements (discussed in Section 3.2.1).

### 2.3.3 Hygroscopicity correction

To compare the CALIOP-derived ambient extinction coefficients with the results of the dry measurements conducted during ATom, we need to correct the former for the effect of hygroscopic growth. Furthermore, the extinction-coefficient-to-ANC conversion discussed in Section 2.3.2 holds only for dry conditions. The POLIPHON method assumes a constant RH of 80 % for marine and 60 % for continental aerosols and may result in errors for higher RH conditions. MOPSMAP includes an in-built functionality to address hygroscopic growth based on the kappa parametrization (Petters and Kreidenweis, 2007) in

the RH range from 0 % to 99 %. We use the normalized aerosol size distributions and refractive indices of different aerosol types from CALIPSO aerosol model to calculate the extinction coefficient for different values of relative humidity. Figure 1 shows the variation of the hygroscopic growth factor, i.e. the ratio between the ambient and dry extinction coefficient, with relative humidity for continental (polluted continental, clean continental, elevated smoke) and marine CALIPSO aerosol types with kappa values of 0.3 and 0.7, respectively. The kappa values are global averages and are suggested by Andreae and

Rosenfeld (2008) for use in satellite retrievals. Nevertheless, studies have found that the kappa values may vary with the aerosol composition and age (Pringle et al., 2010; Cheung et al., 2020). Thus considering a fixed kappa value for a particular aerosol



type defined in CALIPSO may incur additional uncertainties in the ANC retrieval. Moreover, the RH values included in the CALIPSO level 2 data product are estimated from global model simulations which may incorporate additional uncertainties. Having said that, we still use the parametrization with globally averaged kappa values, which were found to provide reasonable

results in the case study presented in Choudhury and Tesche (2022) and the example cases presented later in Section 3.1. Mineral dust is considered to be hydrophobic in our analysis. For every CALIOP data bin, the extinction coefficient is corrected based on the aerosol type and relative humidity value by dividing it with the hygroscopic growth factor that corresponds to the ambient relative humidity. Note that this methodology is different from the one used in Choudhury and Tesche (2022), where the hygroscopicity correction is applied to the particle size distribution before the computation of the ANC. In the present study,

the application of the hygroscopicity correction to the extinction coefficient is necessary so that the dry extinction coefficient from the CALIOP measurements can be compared directly with the ATom data set. The hygroscopicity corrected extinction coefficient is then used to compute the CALIOP-based ANC using the OMCAM and POLIPHON algorithms. Note that in the case of POLIPHON, we only apply the hygroscopicity correction when RH is greater than 80 % and 60 % for marine and continental aerosols, respectively, and modify the corresponding ambient extinction coefficient to RH values of 80 % and 60

200    %. This is because the extinction to ANC conversion equations (Eq. 1) was formulated assuming such RH values which are representative of typical marine and continental environments.

### 2.4   Data matching and comparison

The ATom data consists of continuous airborne in-situ measurements from altitudes of 200 m to 12 km. The measurement tracks for the first ATom campaign are shown in Figure 2. For a comparison between the ANC derived from CALIOP obser-

vations and airborne in-situ measurements conducted during ATom, we need to find those cases for which the two data sets are closest in time and space. In our first attempt at finding intercepts between the tracks from CALIPSO and ATom, we did not consider the aircraft flight level and matched only the 2d latitude and longitude coordinates. As a result, we found that most of the intercepts were found at altitudes above 5 km within the free troposphere. At such altitudes, CALIOP rarely detects aerosol structures except for elevated layers from long-range transport. Hence, we limit the ATom data in the present study to altitudes

below 5 km before finding intercepts with the CALIPSO ground track. This slicing results in a collection of discontinuous measurements during either ascent or descent, or both (v-shaped). Such segments have a latitudinal extent of about 1 to 2 degrees which facilitates the incorporation of the HYSPLIT air-parcel trajectory model (Draxler and Rolph, 2010) for finding the intercepts.

      Major parts of the ATom measurements were conducted over the Pacific and Atlantic oceans. Compared to over land, the

aerosol composition over the oceans is rather homogenous and we can expect a good correlation between ground-based and satellite measurement (Kovacs, 2006; Liu et al., 2008; Tesche et al., 2013). Therefore, we include the CALIPSO tracks that are within 500 km from an ATom measurement in our comparison. Also, for smaller distances, the airborne measurements should be appropriately connected to the nearby CALIPSO overpass. We use HYSPLIT air-parcel trajectories to first determine the section of the CALIPSO overpass that is most appropriate for comparison with the ATom measurements and to second

estimate the correct temporal difference between the measurements. This approach is also used in (Tesche et al., 2013, 2014)

for validating CALIPSO measurements against ground-based lidar and in-situ measurements. For running HYSPLIT, we use Global Data Assimilation System (GDAS) meteorological files with a spatial and temporal resolution of 1 degree and 3 hours, respectively. The overall track selection methodology is illustrated in Figure 3 for an ATom1 flight segment on 8 May 2016. Since the flight measurements are three dimensional, each of the HYSPLIT initialization coordinates has a unique combination

of latitude, longitude, and altitude. To reduce the complexity of the analysis, we limit the initial trajectory starting points by selecting one out of every 20 points in the segment of the aircraft track. Figure 3a shows the forward and backward trajectories starting and ending at different altitudes of the ATom track segment, respectively, and the segment of CALIPSO measurements that is most suitable for the comparison. The vertical displacement of the air parcels along the trajectories is shown in Figure 3b. For most of the found intercepts, the vertical displacement of the air parcels along the trajectories is negligible and, hence,

not considered in our comparison study. As seen in Figure 3a, the trajectories intercept the CALIPSO track at different times. In such situations, we compute the net time difference by averaging the time differences at different height levels. For the example shown in Figure 3a, the air parcels take 9 h (between 1 and 3 km) to 13 h (below 1 km) to reach the CALIPSO track, which leads us to apply an average time delay of 11 h. Including the pre-existing time delay of approx. 9 h between the two observations, the average effective time difference for this case is 2 hours. The average distance between the two tracks as

calculated using the Haversine formula is found to be 457 km. Following this approach, we identified a total of 53 intercepts for which the measurements of CALIOP and ATom are considered as appropriate for comparison. A detailed overview of these cases is given in Table A1 along with the aerosol-type specific extinction coefficient contribution and the average distance and time delay between the observations. The average distance between the tracks is less than 500 km for all the intercepts. The time delay between the measurements varies from 0 to 20 hours with 11 cases exceeding 10 hours. Marine aerosols are found

to be the dominant aerosol type in 44 cases (83 %), followed by polluted continental (4 cases), elevated smoke (3 cases), and dust (2 cases). Such conditions are not unexpected as most of the observations are over oceans. Note that there were many further intercepts where factors like signal attenuation due to the presence of clouds, low signal-to-noise ratio due to low aerosol concentrations or an absence of aerosols lead to CALIOP data that were not suitable for comparison with the ATom measurements. Most of these intercepts were found close to the poles and the equator.

The atmospheric parameters included in the ATom data are at standard temperature (273.15 k) and pressure (1013.25 hPa) and need to be converted to ambient conditions. The temporal resolution of ATom data used in this work is 10 seconds and the corresponding altitudinal resolution varies between 0 and 110 m depending on the speed of the aircraft. However, the vertical resolution of CALIOP data is 60 m in the troposphere. Also, there can be more than one measurement for a certain altitude range in an ATom segment as it can include both ascending and descending measurements. To compare the two data sets, we

thus re-grid them to a uniform vertical resolution of 240 m (4 CALIOP height bins) between 0 and 5 km altitude by averaging both data sets within these height bins. This approach also compensates for the potential vertical displacement of air parcels along the trajectory between the locations of the measurements of CALIOP and the ATom aircraft. However, a limitation to this methodology is the velocity shear at different height levels. It is worthwhile to note that the main motive of this study is to validate the ANC as retrieved from CALIOP data rather than the extinction coefficient. Even after considering all the complex

screening constraints aimed at identifying the best match between CALIOP and ATom measurements by compensating the





temporal and spatial differences between them, disagreement may still arise because of different (i) measuring instruments with dissimilar sensitivities used in ATom and CALIPSO, (ii) measurement techniques, and (iii) spatial and temporal resolutions of the datasets (Tesche et al., 2014). The extinction coefficient from ATom is obtained by applying the Mie theory to the dry aerosol size distributions for radius <2.4 μm. This may be inaccurate for coarse mode non-spherical aerosol particles. The

260 CALIPSO retrievals on the other hand have to go through a complex feature detection algorithm to identify aerosol layers and may fail to detect optically thin layers with inadequate signal to noise ratio. While the airborne in-situ data from ATom are point measurements, the along-swath width of the CALIPSO level 2 data bin is 5 km. Moreover, the HYSPLIT trajectories used to find the intercepts use model outputs and may have associated errors. Even so, it is necessary to perform a closure study utilizing these concurrent measurements for validating the recently developed lidar-based ANC retrieval algorithms. In order

to somewhat compensate for such unquantifiable effects in the comparison of ANC, we only use those data bins for which the difference between the dry extinction coefficient from CALIOP is within ±50 % of that in the ATom data. This additional filter further increases the probability that we are comparing the ANC within the same air parcel.

## 3 Results

### 3.1 Example cases

We start the presentation of results in Figure 4 with four comparison examples that present the profiles of extinction coefficient and ANC as derived from ATom and CALIOP measurements. The first three cases represent different prevailing aerosol types while the fourth shows a combination of all four types. The majority of cases includes airborne measurements during both ascent and descent and, hence, there can be two ATom measurements at one level. All CALIPSO overpasses except for the marine dominated case shown in the examples occurred during nighttime.

The first example case for the CALIPSO and ATom measurement on 15 Feb 2017 is shown in Figures 4a & 4e. The case is dominated by the presence of marine aerosols with 85 % of the CALIPSO bins below 1 km having RH>80 %. Close to the surface (below 300 m), the RH exceeded 99 % due to which a finite dry extinction coefficient could not be retrieved. However, for altitudes higher than 300 m, we found a reasonable agreement between the humidity corrected extinction coefficient from CALIOP and the ATom measurements (Figure 4a). This illustrates the ability of the kappa parametrization to account for

aerosol hygroscopicity for highly humid marine environments. The $n_{50,dry}$ profiles derived from CALIOP data using the POLIPHON technique is at par with that measured during ATom. However, the OMCAM estimates are relatively noisy, perhaps because of highly variable RH, and are lower than the ATom measurements for most altitudes. This is also evident in other marine-dominated cases e.g., near-surface measurements in Figure 4h. However, in the case of $n_{250,dry}$, both the OMCAM and POLIPHON estimates for marine-dominated CALIPSO retrievals are in much better agreement with the ATom data.

The second example of the intercept on 17 August 2016 is dominated by a mixture of marine and smoke aerosols at altitudes below 1.5 km and only smoke at higher altitudes. Figure 4b shows that the extinction coefficients from CALIOP and ATom are at par below 2 km altitude. At higher altitudes, where elevated smoke is the dominant aerosol type, CALIOP gives much higher extinction coefficients than found from the ATom measurements. A plausible reason behind the larger values is perhaps





the temporal (11 h) and spatial (205 km) difference between the observations. The properties of an elevated smoke layer may
change drastically with the travelled distance and age of the air parcel. Though the CALIOP-derived $n_{50,\text{dry}}$ and $n_{250,\text{dry}}$
profiles using POLIPHON and OMCAM accurately capture the altitudinal variation revealed in the ATom measurements, they
are far more variable with altitude and differ from the in-situ measurements at altitudes between 2 km and 4 km.

In the third example of 1 October 2017, the aerosol types detected by the CALIPSO retrieval are polluted continental and
mineral dust, with the former dominating. The CALIOP extinction coefficient and $n_{50,\text{dry}}$ are in good agreement with the
295 ATom measurements. However, the $n_{250,\text{dry}}$ (Figure 4g) as estimated from CALIOP using both the OMCAM and POLIPHON
algorithms is 2 to 5 times larger than in the ATom measurements. On analyzing the geographical locations of the measurements,
we found that both of them are over land regions (Southern California) and encompass a mixture of urban, rural, and forest
continental environments. The aerosol properties can be highly variable over different land regions which perhaps is the reason
behind the disagreement of the $n_{250,\text{dry}}$ values.

The fourth example for the intercept on 29 April 2018 is comprised of a mixture of all four aerosol types with marine aerosols
dominating from the surface to 1 km, followed by continental and smoke aerosols until 3 km, and further accompanied by min-
eral dust over 3 km (Figure 4d). The ATom-derived extinction coefficient (for ascending and descending flight-track segments)
varies by as much as 1.5 orders of magnitude at heights above 2 km. This highlights the impact of spatial heterogeneity that
may occur over short distances or time periods. The CALIOP-derived humidity-corrected extinction coefficient resembles the
305 in-situ measurements during ascent (with larger values than during descent) between 1 and 4 km altitude. Above and be-
low that layer, the CALIOP extinction coefficient exceeds that derived from the in-situ measurements. Regarding $n_{50,\text{dry}}$, the
POLIPHON estimate overlaps with the ATom measurements up to an altitude of 4 km, above which it fails to replicate the
increase in aerosol concentration. The OMCAM-derived profile in Figure 4h shows a similar agreement but underestimates
$n_{50,\text{dry}}$ at altitudes below 1 km where marine aerosols are dominant. The $n_{250,\text{dry}}$ as estimated from POLIPHON and OMCAM
are both in reasonable agreement with the ATom measurements.

Overall, the example cases in Figure 4 present a remarkable resemblance of the aerosol properties derived from CALIOP
observations with the ATom measurements at most height levels. The examples that feature dominance of marine aerosols in
the lowermost 2 km illustrate the importance of applying a hygroscopicity correction and indicate that this can be realized
to a reasonable degree with the kappa parametrization even when using static kappa values. In the next section, we present a
315 statistical comparison of the extinction coefficient and ANC for all the identified intercepts.

## 3.2 General findings

Figure 5 presents a comparison of the aerosol extinction coefficient as derived from ATom and CALIOP measurements for
all the identified intercepts and with data re-gridded to a unified altitude profile with 240 m bin width. The correlation of the
data sets gives a Spearman's correlation coefficient (R) value of 0.715, a root mean square error (RMSE) of 0.017 $\text{km}^{-1}$,
and a bias of -0.007 $\text{km}^{-1}$ (Figure 5a). For the aerosol-type specific comparison, individual height bins were separated based
on the dominant aerosol type, i.e. the one which shows the largest extinction coefficient. In terms of correlation coefficient,
best agreement is found for polluted continental aerosols (R=0.805), followed by marine (R=0.744), mineral dust (R=0.583),





and smoke (R=0.4). A similar level of agreement is also seen in terms of the RMSE and bias values given in Figure 5b-e. Moreover, both data sets are in better agreement at altitudes below 2 km irrespective of the dominant aerosol type. Such a
325 result is expected as elevated aerosols above the boundary layer can be easily transported to larger distances compared to those located near the surface which counteracts the comparison approach followed in this study.

As seen from the general comparison and case studies, the aerosol extinction coefficient inferred from ATom measurement is in very good agreement with the CALIPSO retrieval with the exception of a few cases where they can be significantly different. Scenarios that may lead to large differences in the data sets are already discussed in Section 2.4 and includes the differences
in the instrument sensitivities, measurement techniques, spatial and temporal resolutions, and assumptions underlying the intercept identification. In such situations, comparing the corresponding ANC may lead to misleading conclusions. Thus, while comparing the ANC, we only use those altitude bins for which the CALIOP-derived dry extinction coefficient is within $\pm 50\,\%$ of that estimated from ATom measurement. Note that the present study is not focused on the evaluation of CALIPSO products, for which several studies have already been performed (Mamouri et al., 2009; Pappalardo et al., 2010; Omar et al., 2013;
Tesche et al., 2013, 2014; Kacenelenbogen et al., 2014; Rogers et al., 2014; Papagiannopoulos et al., 2016). By introducing the additional constraint of a set difference in the extinction coefficient we aim to further increase the likelihood of comparing the same air parcels.

The comparison of $n_{50,\mathrm{dry}}$ as measured during ATom and estimated from CALIOP measurements using OMCAM and POLIPHON for the altitude bins that pass the extinction coefficient filter is shown in Figure 6. It is found that the POLIPHON
estimates of $n_{50,\mathrm{dry}}$ are in better agreement with the ATom measurements with a correlation coefficient of 0.829, RMSE value of 234 $\mathrm{cm}^{-3}$, and bias value of -96.627 $\mathrm{cm}^{-3}$. In terms of absolute magnitude, OMCAM estimated $n_{50,\mathrm{dry}}$ are up to an order less than that of ATom, especially for aerosol concentrations below 100 $\mathrm{cm}^{-3}$. A closer look at the aerosol-type specific comparison shows that the lower values seen in OMCAM is primarily from the marine-dominated cases for which POLIPHON estimates of $n_{50,\mathrm{dry}}$ are generally in better agreement with the in-situ measurements. For dust-dominated cases,
both the algorithms perform similar, with POLIPHON being slightly better in terms of R, bias and RMSE values. However, the POLIPHON-derived values for dust aerosols are $n_{100,\mathrm{dry}}$ instead of $n_{50,\mathrm{dry}}$, and thus, should in principle, underestimate the $n_{50,\mathrm{dry}}$. POLIPHON underestimates the $n_{50,\mathrm{dry}}$ for approx. 37 % retrievals. Given the limited sample space (19 bins), at the current stage, it is hard to comment on the performance of POLIPHON for dust dominated cases. For the cases where polluted continental aerosols are dominant, the $n_{50,\mathrm{dry}}$ as estimated from both the algorithms are in good agreement with the
ATom in-situ measurements. Statistically speaking, OMCAM (R=0.609, RMSE=275.93, bias=26.548) has better agreement with the ATom data than POLIPHON (R=0.457, RMSE=335.81, bias=-125.757). A similar result is also found for cases dominated by elevated smoke for which both the POLIPHON (R=0.658, bias=-171.491, RMSE=308.46) and OMCAM (R=0.791, bias=105.47, RMSE=213.33) estimates of $n_{50,\mathrm{dry}}$ are in very good agreement with the ATom measurements. Interestingly, after applying the extinction-coefficient constraint for comparing both the datasets, the CALIOP-estimated $n_{50,\mathrm{dry}}$ values are
in good agreement with the ATom measurements even at higher altitudes.

Figure 7 depicts the comparison of $n_{250,\mathrm{dry}}$ as derived from ATom and CALIOP measurements for the altitude bins that pass the extinction-coefficient constraint. From the figure, we find that both the OMCAM and POLIPHON derived $n_{250,\mathrm{dry}}$





are in good agreement with the in-situ measurements in terms of the correlation coefficient, RMSE, and bias magnitude. Further, the type-specific comparison shows that for marine-dominated cases, both the algorithms yield similar results and 360 show similar level of agreement with the ATom estimates. For dust-dominated cases, POLIPHON (R=0.525, bias=2.939, RMSE=4.22) estimated $n_{250,\mathrm{dry}}$ values are in marginally better agreement with the ATom data than OMCAM (R=0.468, bias=3.439, RMSE=4.61). For polluted continental and elevated smoke dominant cases, the $n_{250,\mathrm{dry}}$ estimated from OMCAM and POLIPHON algorithms show similar agreement with the corresponding ATom measurements.

### 3.2.1 Revised OMCAM algorithm

Figures 6b revealed that CALIOP-derived $n_{50,\mathrm{dry}}$ from the OMCAM algorithm for marine-dominated cases resulted in smaller values compared to that from POLIPHON and in-situ measurements. In this section, we estimate ANC from CALIOP using a revised OMCAM algorithm in which a marine model derived from 11 AERONET island stations (Sayer et al., 2012) is used to characterize the marine aerosols. This new marine model is used to correct the CALIOP measurements for hygroscopicity by estimating the growth factors at different RH values (Figure 1). Also, the conversion factors for $n_{50,\mathrm{dry}}$ and $n_{250,\mathrm{dry}}$ are 370 re-calculated (Table 3) using the updated marine model following the methodology discussed in Section 2.3.2. It is interesting to note that the conversion factor estimated from the new marine model for $n_{250,\mathrm{dry}}$ is only increased by 5 %, compared to 520 % for $n_{50,\mathrm{dry}}$. For comparing the CALIOP and Atom measurements for all the identified intersections, we only use those 240 m data bins that pass the extinction-coefficient constraint (CALIOP-RH-corrected extinction coefficient within ±50 % of ATom measurement). Figure 8 depicts the comparison of $n_{50,\mathrm{dry}}$ and $n_{250,\mathrm{dry}}$ as derived from ATom and inferred from CALIOP 375 data using the revised OMCAM algorithm. The figure shows that both the OMCAM estimates of $n_{50,\mathrm{dry}}$ and $n_{250,\mathrm{dry}}$ are in very good agreement with the in-situ measurements when resorting to the marine model of Sayer et al. (2012) with the $n_{50,\mathrm{dry}}$ comparison giving a correlation coefficient of 0.791, a RMSE of 135.42 $\mathrm{cm}^{-3}$, and a bias of -21.68 $\mathrm{cm}^{-3}$. The estimates of ANC from the updated OMCAM algorithm for marine aerosol type is now at par with that from POLIPHON.

## 4 Discussion

In general, when the RH corrected extinction coefficient from CALIOP is used, both the OMCAM and POLIPHON algorithms yield values of $n_{50,\mathrm{dry}}$ and $n_{250,\mathrm{dry}}$ that are comparable to in-situ measurements for all aerosol types except for marine-dominated cases. For marine-dominated retrievals, even though the $n_{250,\mathrm{dry}}$ estimated from OMCAM and POLIPHON algorithms were in good agreement with the in-situ measurements, OMCAM estimates of $n_{50,\mathrm{dry}}$ were up to an order smaller. This is perhaps the result of the limited in-situ sea salt size distribution measurements that form the marine aerosol model used in 385 the OMCAM algorithm (Omar et al., 2009; Choudhury and Tesche, 2022). Nevertheless, using the AERONET based marine model of (Sayer et al., 2012) in OMCAM results in an overall better agreement for both the $n_{50,\mathrm{dry}}$ and $n_{250,\mathrm{dry}}$ values with the independent airborne in-situ measurements during ATom.

For dust-dominated retrievals, we find a moderate correlation between CALIOP-derived results and the in-situ measurements. For both the $n_{50,\mathrm{dry}}$ and $n_{250,\mathrm{dry}}$, POLIPHON gives a marginally better agreement with the in-situ data. Still, the





POLIPHON conversion factors for mineral dust relate to $n_{100,\mathrm{dry}}$ and not to $n_{50,\mathrm{dry}}$. For some cases, the ANC estimated from both the algorithms are significantly different from the in-situ measurements. Also, both the algorithms result in lower $n_{250,\mathrm{dry}}$ values compared to the in-situ data for most cases contrary to the results from Haarig et al. (2019) that reports an excellent agreement between ground-based lidar and air-borne in-situ measurements taken during the SALTRACE campaign at Barbados. On further investigating the locations of dust dominated intersections, we found that the underestimation is independent

of the geographic location and is evident for retrievals over both the Atlantic and Pacific oceans. The aerosol type identified by CALIPSO for dust-dominated cases is mostly dusty marine (dust + marine) and not pure dust. Under such situations where the dust particles are far away from their source regions, their microphysical properties may change because of either ageing or chemical or cloud processing (Kim and Park, 2012; Ansmann et al., 2019; Goel et al., 2020). Also, three-fourths of the CALIPSO retrieval for dust-dominated cases are daytime retrievals. This might add to the differences observed between the

observations.

For retrievals dominated by polluted continental and smoke, we find a medium-high correlation between ATom measurements and CALIOP-inferred estimates of $n_{50,\mathrm{dry}}$ using both algorithms with OMCAM performing slightly better than POLIPHON. For some height bins, the CALIOP estimates vary by more than a factor of 2 (especially for $n_{250,\mathrm{dry}}$) from the in-situ measurements. Such a variation may either occur because of the spatial and temporal heterogeneity of aerosols or due

to change in the microphysical properties of the aerosols as a result of chemical or cloud processing. Also, similar to dust aerosols, the conversion factors for smoke and continental aerosols may change with age and geographical location (Ansmann et al., 2021).

Overall, the $n_{50,\mathrm{dry}}$ and $n_{250,\mathrm{dry}}$ values estimated from the OMCAM algorithm with the updated marine model and the POLIPHON algorithm are in overall good agreement with the ATom in-situ measurements. Such concurrence between the

410 satellite estimates of height resolved ANC (that are most relevant for cloud processes) and the coincident in-situ measurements for various aerosol environments has not been achieved yet. This study along with previous concurrent results (Haarig et al., 2019; Marinou et al., 2019; Georgoulias et al., 2020; Choudhury and Tesche, 2022) compliments the use of ground-based and spaceborne lidar remote sensing techniques for retrieving height-resolved cloud-relevant aerosol microphysical properties.

## 5   Summary

We present a validation study of the spaceborne lidar derived aerosol number concentration using the OMCAM and POLIPHON algorithms with the airborne in-situ measurements conducted during the ATom campaigns over the Atlantic and Pacific oceans. To identify the comparison cases, we located intercepts between the CALIPSO flight tracks and the ATom aircraft tracks with the help of HYSPLIT trajectories. Out of all intercepts, 53 were found to be suitable for comparison. On comparing the dry extinction coefficients, we found an overall good agreement between the CALIOP data and the in-situ measurements with a

correlation coefficient of 0.715. Disagreement was found mostly for retrievals above 3 km altitude. Such differences are most likely due to the spatial heterogeneity of aerosol properties rather than a retrieval error. Therefore, to compare the ANC, we filtered the data sets to select only those retrievals for which the CALIOP extinction coefficient is within ±50 % of the one

obtained from the in-situ measurements. This constraint further increases the likelihood of comparing the same air parcel, which is crucial for parameters such as ANC that can easily vary by many orders of magnitude in space and time.

We found that the POLIPHON and OMCAM estimates of $n_{50,\text{dry}}$ are in overall good agreement with the in-situ measurements with an overall correlation coefficient of 0.829 and 0.823, respectively. The agreement is seen for all the dominating aerosol type with the exception of marine aerosols, for which the POLIPHON estimates give a better agreement than the OMCAM. Revising OMCAM with the marine model of Sayer et al. (2012) lead to results similar to the ones from POLIPHON and an overall better agreement with the in-situ measurements. In the case of $n_{250,\text{dry}}$, it is found that both the OMCAM (R=0.463) and POLIPHON (R=0.47) are in reasonable agreement with the in-situ measurements. The updated OMCAM algorithm for marine aerosols resulted in no significant change in the $n_{250,\text{dry}}$ concentrations.

Given the importance of knowledge regarding the global 3D distribution of the concentration of cloud-relevant aerosol particles, both the POLIPHON and the OMCAM (with the revised treatment of marine aerosols) algorithms emerge as an effective way for estimating the aerosol number concentrations over different size ranges from spaceborne lidar measurements. The next step is to test the potential of CALIOP measurements in deriving CCN concentrations by validating them with long-term surface in-situ measurements, for example from (Schmale et al., 2017). The best performing algorithm will be used to derive global CCN climatology from CALIPSO. This dataset will be beneficial in evaluating models and other satellite products, region and regime wise detailed ACI studies, and for better constraining aerosol radiative forcing estimates in climate models.

*Data availability.* The CALIPSO level 2 v4.20 aerosol profile data product used in this work is available at https://doi.org/10.5067/CALIOP/ CALIPSO/LID_L2_05KMAPRO-STANDARD-V4-20 (CALIPSO, 2018). The v1.5 of ATom: Merged Atmospheric Chemistry, Trace Gases, and Aerosols data is available at https://doi.org/10.3334/ORNLDAAC/1581 (Wofsy et al., 2018).

*Author contributions.* GC conceptualized the study, performed the data analysis, and prepared the plots under the guidance of MT. GC and MT prepared the initial version of the manuscript. GC, MT, and AA contributed to the discussion of the findings and the revisions of the manuscript.

*Competing interests.* The authors declare that they do not have any competing interests.

*Acknowledgements.* We acknowledge the NOAA Air Resources Laboratory (ARL) for providing the HYSPLIT transport and dispersion model and/or READY website (http://www.ready.noaa.gov, last access: 22 February 2022) used in this manuscript. We thank the CALIPSO science team for providing the CALIPSO data. We thank the AERIS/ICARE Data and Services Center for providing access to the CALIPSO





data used in this study. We gratefully acknowledge the Oak Ridge National Laboratory Distributed Active Archive Center (ORNL DAAC) for maintaining and openly sharing the ATom data as well as the PI's, technical, and non-technical members involved in the ATom campaigns.

*Financial support.* This research has been supported by the Franco-German Fellowship Programme on Climate, Energy, and Earth System Research (Make Our Planet Great Again – German Research Initiative, MOPGA-GRI, grant number 57429422) of the German Academic Exchange Service (DAAD), funded by the German Ministry of Education and Research.



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



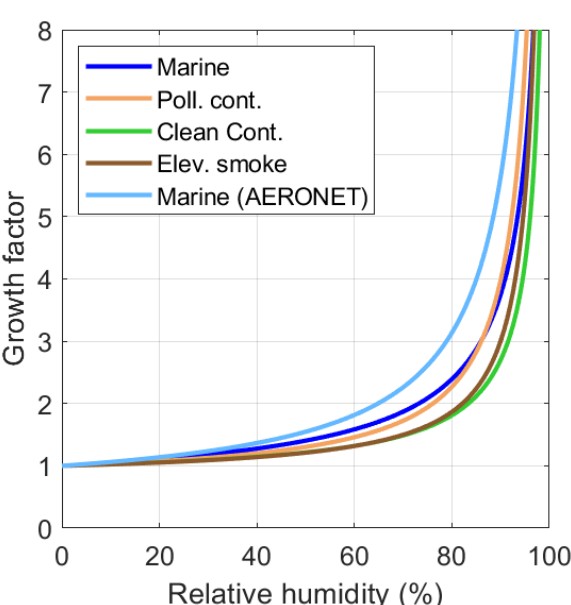

**Figure 1.** Hygroscopic growth factor for different values of relative humidity for different aerosol types estimated from the microphysical properties of CALIPSO aerosol models: marine (blue), polluted continental (orange), clean continental (green), and elevated smoke (brown). The growth factor estimated using a new AERONET based marine model (sky blue) from Sayer et al. (2012) is also shown. The hygroscopic growth factor at a certain relative humidity is defined as the ambient to dry extinction coefficient ratio.

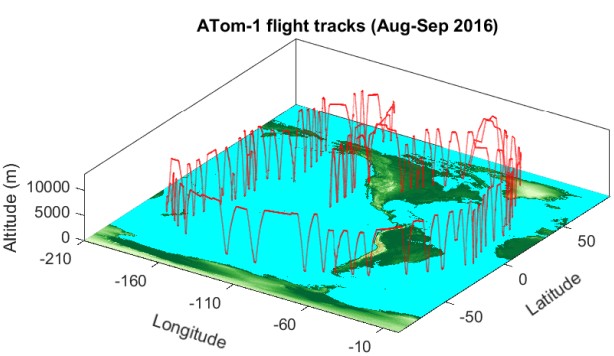

**Figure 2.** Flight tracks during the ATom1 campaign (red lines) carried out from August to September 2016 covering altitudes from 200 m to 12 km over the Pacific and Atlantic oceans.



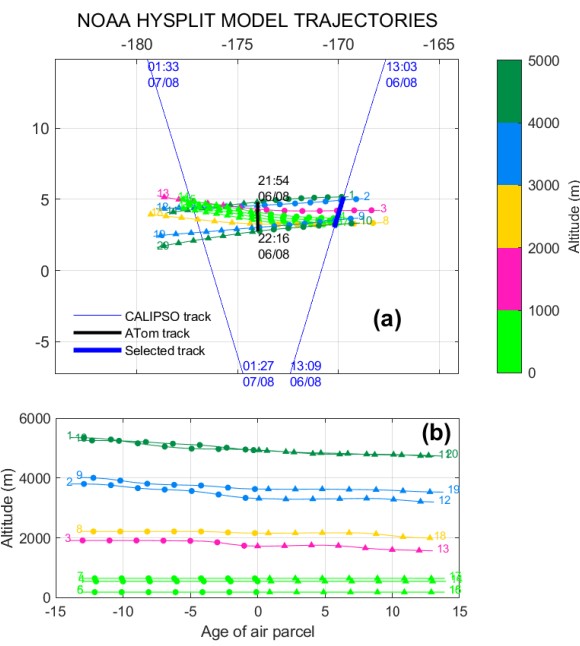

**Figure 3.** (a) CALIPSO overpasses (dark blue lines) close to the ATom measurements on 6 August 2016 with HYSPLIT backward (lines with filled circles) and forward (lines with filled triangles) trajectories starting and ending at different points along the ATom track. The color bar represents the altitude of ATom coordinates used to compute the HYSPLIT trajectories. The CALIPSO track selected for comparison is highlighted as bold blue line in the CALIPSO overpass at 13:04 UTC on 6 August 2016. (b) Vertical displacement of the air parcels along the individual HYSPLIT trajectories. Each track is associated with a number to identify its vertical displacement and time difference.

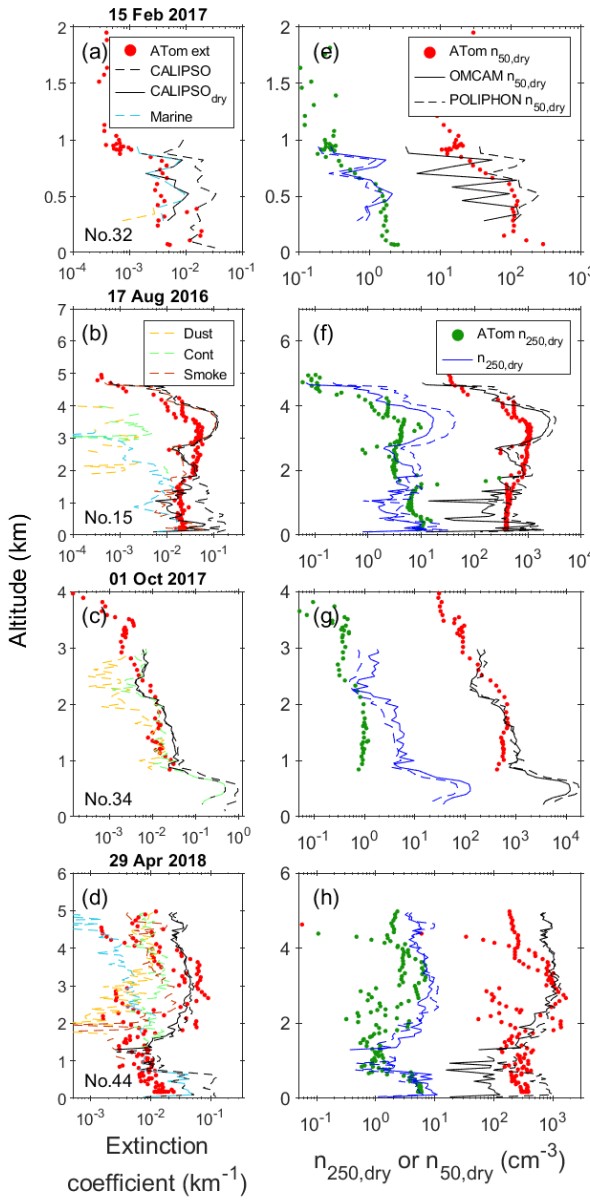

**Figure 4.** Profiles of aerosol extinction coefficient at 532 nm (a-d), aerosol number concentration (ANC) with dry radius >50 nm (RHS of e-h) and with dry radius >250 nm (LHS of e-h) retrieved from ATom and CALIPSO measurements for four selected cases (each in one row). The dashed and solid black lines in plots a-d denote CALIPSO derived ambient and RH corrected extinction coefficient, respectively. The dashed coloured lines in a-d refers to RH corrected extinction coefficients of individual aerosol types. The solid and dashed lines in e-h refers to the ANC derived using OMCAM and POLIPHON techniques, respectively. The serial number of cases in Table A1 is also given in the lower left corner of the plots.



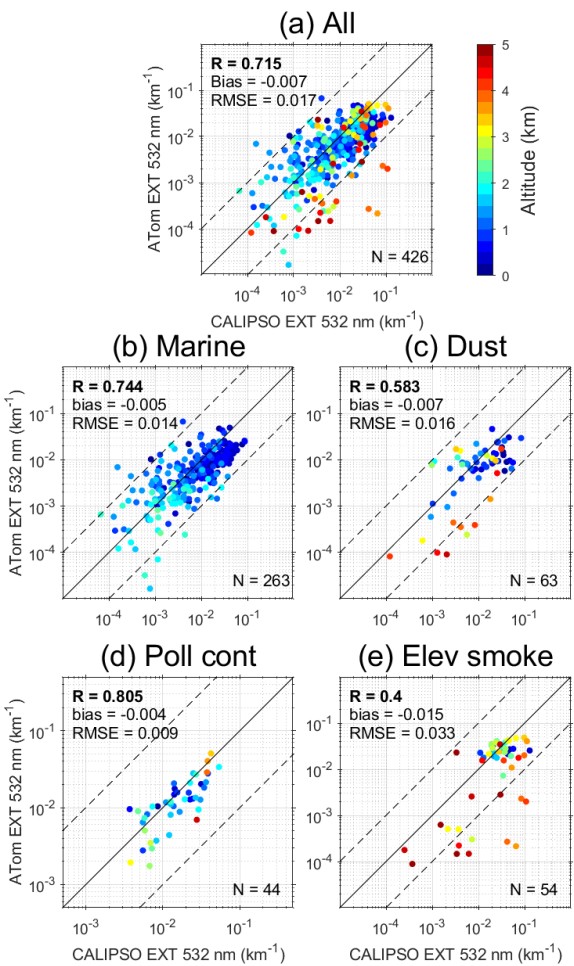

**Figure 5.** Comparison of dry aerosol extinction coefficient from CALIOP observations and ATom measurements between the surface and 5 km altitude with data re-gridded to a profile with 240 m bin width and colors referring to the altitude of the measurement (a). The bins where marine (b), mineral dust (c), polluted continental (d), and elevated smoke (e) aerosols are dominant are shown separately. The Spearman's correlation coefficient (R), bias, root mean squared error (RMSE), and the sample space (N) are given in the legend. The solid line represent the identity line and the dotted lines on either side of it represent one order of magnitude from the identity line.



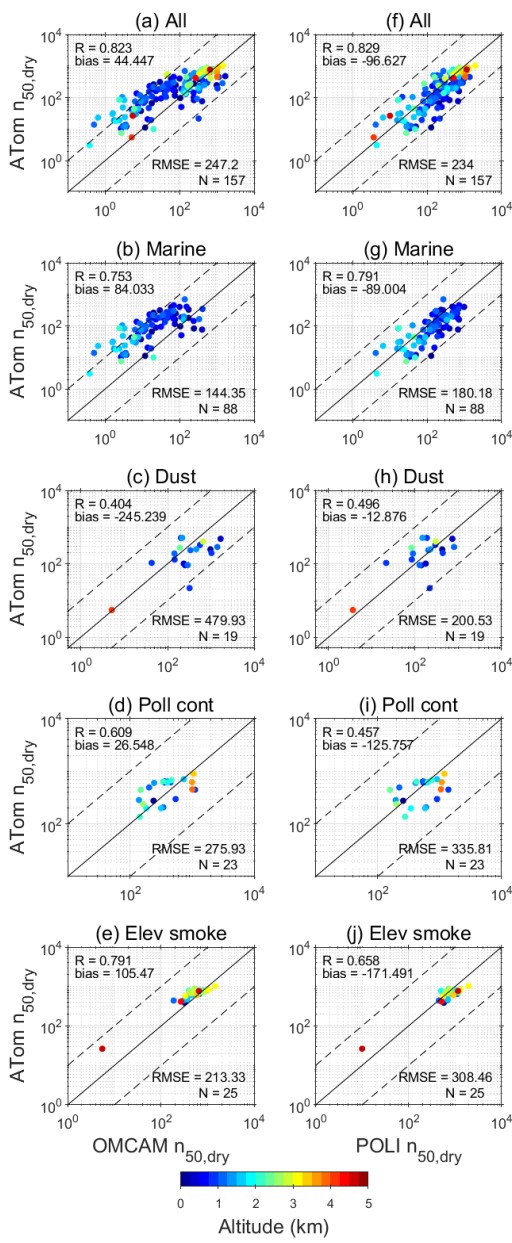

**Figure 6.** Comparison of $n_{50,\mathrm{dry}}$ (in $\mathrm{cm}^{-3}$) retrieved from ATom and CALIPSO measurements using OMCAM (a-e) and POLIPHON (f-j) for 240 m altitude bins between 0 and 5 km for all the identified intersections. The bins where marine (b & g), dust (c & h), polluted continental (d & i), and elevated smoke (e & j) aerosols are dominant are separately shown. The Spearman's correlation coefficient (R), bias, root mean squared error (RMSE), and the sample space (N) are given in the legend.



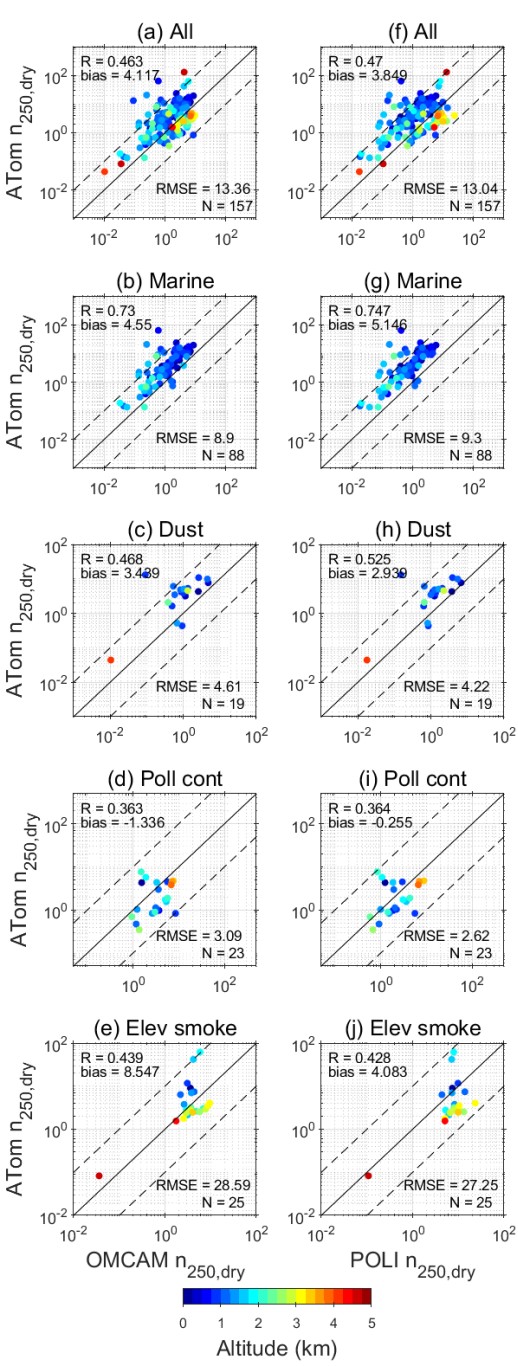

**Figure 7.** Same as Figure 6 but for $n_{250,\mathrm{dry}}$.





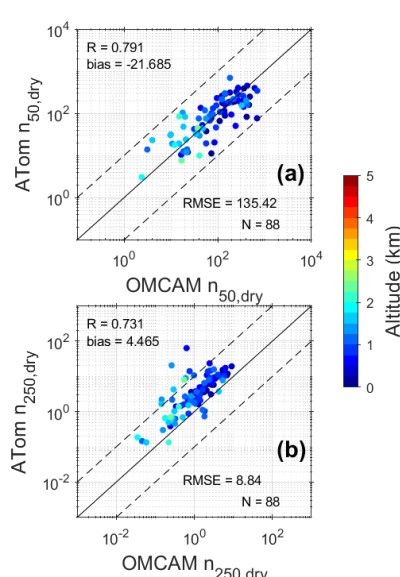

**Figure 8.** $n_{50,\mathrm{dry}}$ (a) and $n_{250,\mathrm{dry}}$ (b) derived from ATom and CALIOP using the updated OMCAM algorithm for marine-dominated altitude bins.



**Table 1.** Details of the ATom parameters used in the comparison study.

| Instrument | Parameter name | Description |
|---|---|---|
| Laser aerosol spectrometer (LAS) | Nacc_LAS | Number concentration of dry aerosols for ammonium sulfate optical equivalent radius (R) = 0.05 to 0.44 μm at STP |
| | Ncoarse_LAS | Number concentration of dry aerosols for 0.44≤R≤2 μm at STP |
| Nucleation-mode aerosol size spectrometer (NMASS), ultra-high sensitivity aerosol size spectrometer (UHSAS), LAS | calc_ext_532_AMP | Total calculated particle extinction at 532 nm wavelength assuming dry ammonium sulfate for 0.00135≤R≤2.4 μm at STP |



**Table 2.** POLIPHON conversion factors and extinction exponents for different aerosol types to be used in Eq. 1 for estimating aerosol number concentrations from extinction coefficient. The value of extinction exponent ($x$) for $n_{250,\mathrm{dry}}$ is 1 for all the aerosol types.

| Type | $n_{50,\mathrm{dry}}$ ($n_{100,\mathrm{dry}}$ for dust) | | $n_{250,\mathrm{dry}}$ | Source |
|---|---|---|---|---|
| | Conversion factor (Mm cm$^{-3}$) | Extinction exponent ($x$) | Conversion factor (Mm cm$^{-3}$) | |
| Dust | 8.855 | 0.7525 | 0.1475 | Ansmann et al. (2019) |
| Continental | 25.3 | 0.94 | 0.1 | Mamouri and Ansmann (2016) |
| Marine | 7.2 | 0.85 | 0.06 | Mamouri and Ansmann (2016) |
| Smoke | 17 | 0.79 | 0.35 | Ansmann et al. (2021) |





**Table 3.** OMCAM conversion factors to calculate ANC from Eq. 3.

| Type | Conversion factors (Mm cm$^{-3}$) | |
|---|---|---|
| | $n_{50,\mathrm{dry}}$ ($n_{100,\mathrm{dry}}$) | $n_{250,\mathrm{dry}}$ |
| Dust | 42.9728 (11.0847) | 0.0865 |
| Clean continental | 3.598 | 0.1995 |
| Polluted continental | 24.931 | 0.2601 |
| Smoke | 21.9948 | 0.1446 |
| Marine | 2.3988 | 0.2084 |
| Modified marine | 14.8569 | 0.2184 |



Table A1: Details of ATom and CALIPSO data for the identified intersections. Δt is the effective time difference between the tracks after incorporating HYSPLIT trajectories. The difference in distance (Δs) and Δt between the measurements are averaged values. Abbreviations: D-dust, M-marine, C-polluted continental, and S-elevated smoke.

| S. No. | ATom | | | CALIPSO | | | Extinction coefficient contribution (%) | | | | Δs (km) | Δt (h) |
|---|---|---|---|---|---|---|---|---|---|---|---|---|
| | Date | Time | Latitude | Date | Time | Latitude | D | M | C | S | | |
| 1 | 29/07/16 | 1655-1719 | 17.27, 19.43 | 29/07/16 | 2126 (D) | 17.05, 20.99 | 39 | 44 | 17 | 0 | 257 | 16.5 |
| 2 | 01/08/16 | 2322-2347 | 64.35, 65.89 | 01/08/16 | 2210 (D) | 63.22, 64.96 | 0 | 0 | 100 | 0 | 237 | 3.5 |
| 3 | 04/08/16 | 0025-0043 | 30.01, 31.39 | 03/08/16 | 1227 (N) | 29.52, 31.48 | 0 | 100 | 0 | 0 | 126 | 7 |
| 4 | 06/08/16 | 1757-1808 | 18.96, 19.72 | 06/08/16 | 2355 (D) | 18.54, 20.37 | 76 | 24 | 0 | 0 | 83 | 7 |
| 5 | 06/08/16 | 1902-1915 | 17, 17.48 | 06/08/16 | 1302 (N) | 16.01, 17.98 | 1 | 96 | 3 | 0 | 359 | 14 |
| 6 | 06/08/16 | 1955-2018 | 14.03, 15 | 06/08/16 | 1303 (N) | 13.54, 15.46 | 0 | 100 | 0 | 0 | 241 | 2 |
| 7 | 06/08/16 | 2154-2216 | 2.76, 4.80 | 06/08/16 | 1306 (N) | 3.02, 5.17 | 0 | 100 | 0 | 0 | 457 | 2 |
| 8 | 06/08/16 | 2302-2314 | -0.01, 0.004 | 06/08/16 | 1307 (N) | -0.17, 0.13 | 0 | 100 | 0 | 0 | 31 | 10 |
| 9 | 06/08/16 | 2337-0001 | -0.09, 0.001 | 06/08/16 | 1307 (N) | -0.26, 0.49 | 10 | 90 | 0 | 0 | 446 | 20 |
| 10 | 08/08/16 | 1934-1943 | -15.02, -14.31 | 08/08/16 | 1259 (N) | -15.17, -14.02 | 0 | 100 | 0 | 0 | 44 | 7 |
| 11 | 08/08/16 | 2019-2055 | -22.47, -19.34 | 08/08/16 | 1300 (N) | -22.97, -18.81 | 0 | 100 | 0 | 0 | 143 | 7.5 |
| 12 | 13/08/16 | 0431-0456 | -65.24, -64.98 | 13/08/16 | 0834 (N) | -64.19, -61.04 | 14 | 86 | 0 | 0 | 331 | 2 |
| 13 | 15/08/16 | 1114-1140 | -50.79, -49.54 | 15/08/16 | 1833 (D) | -51.76, -49.23 | 7 | 93 | 0 | 0 | 113 | 9 |
| 14 | 15/08/16 | 1210-1238 | -47.46, -45.96 | 15/08/16 | 0459 (N) | -48.76, -46.52 | 13 | 87 | 0 | 0 | 336 | 0.5 |
| 15 | 17/08/16 | 0855-0919 | -3.03, -0.95 | 17/08/16 | 0255 (N) | -4.86, -0.02 | 1 | 14 | 3 | 81 | 205 | 11 |
| 16 | 17/08/16 | 0956-1017 | 3.48, 5.22 | 17/08/16 | 0255 (N) | -0.28, 2.36 | 2 | 3 | 0 | 95 | 408 | 7 |
| 17 | 22/08/16 | 1748-1819 | 45.54, 46.13 | 22/08/16 | 1906 (D) | 45.23, 47.57 | 18 | 0 | 61 | 0 | 125 | 1 |
| 18 | 22/08/16 | 1831-1851 | 44.88, 45.07 | 22/08/16 | 1906 (D) | 44.61, 46.99 | 10 | 0 | 78 | 0 | 330 | 6 |
| 19 | 26/01/17 | 2148-2212 | 0.91, 2.01 | 26/01/17 | 2140 (D) | 0.04, 1.96 | 19 | 47 | 34 | 0 | 147 | 5 |
| 20 | 26/01/17 | 2249-2311 | 6, 8 | 26/01/17 | 2142 (D) | 6.02, 7.98 | 34 | 66 | 0 | 0 | 77 | 0 |
| 21 | 26/01/17 | 2356-0019 | 13.51, 15.48 | 27/01/17 | 0958 (N) | 13.04, 14.96 | 0 | 100 | 0 | 0 | 144 | 5 |
| 22 | 01/02/17 | 2113-2142 | 55.03, 56.08 | 01/02/17 | 1322 (N) | 55.11, 56.97 | 0 | 74 | 26 | 0 | 84 | 9 |
| 23 | 03/02/17 | 1953-2003 | 16, 16.5 | 04/02/17 | 0013 (D) | 16.45, 17.16 | 14 | 86 | 0 | 0 | 197 | 10 |
| 24 | 03/02/17 | 2220-2245 | 4.31, 5.75 | 03/02/17 | 1324 (N) | 3.81, 6.18 | 0 | 100 | 0 | 0 | 157 | 4 |
| 25 | 06/02/17 | 0241-0305 | -56.05, -54.60 | 06/02/17 | 1412 (N) | -56.06, -53.63 | 0 | 100 | 0 | 0 | 327 | 18 |
| 26 | 10/02/17 | 1854-1901 | -43.77, -43.50 | 10/02/17 | 1344 (N) | -44.99, -44.03 | 0 | 75 | 0 | 25 | 113 | 1 |
| 27 | 10/02/17 | 2323-2349 | -64.71, -64.13 | 11/02/17 | 0937 (N) | -65.26, -64.21 | 0 | 18 | 0 | 82 | 103 | 5 |



**Table A1 continued from previous page**

| 28 | 11/02/17 | 0312-0331 | -59.74, -58.72 | 11/02/17 | 0617 (N) | 60.18, -58.38 | 2 | 98 | 0 | 0 | 75 | 0 |
|---|---|---|---|---|---|---|---|---|---|---|---|---|
| 29 | 13/02/17 | 1238-1301 | -46.76, -45.43 | 13/02/17 | 1756 (D) | -50.99, -47.04 | 44 | 56 | 0 | 0 | 346 | 10.5 |
| 30 | 13/02/17 | 1733-1753 | -20.3, -18.87 | 14/02/17 | 0319 (N) | -20.95, -19.04 | 5 | 93 | 0 | 2 | 443 | 2.5 |
| 31 | 13/02/17 | 1930-2001 | -9.11, -7.75 | 13/02/17 | 1450 (D) | -9.47, -6.52 | 39 | 54 | 6 | 0 | 122 | 10 |
| 32 | 15/02/17 | 1713-1729 | 38.76, 39.2 | 15/02/17 | 1451 (D) | 40.01, 44.98 | 6 | 94 | 0 | 0 | 462 | 10 |
| 33 | 19/02/17 | 1848-1914 | 74.2, 76.09 | 19/02/17 | 1934 (D) | 75.54, 77.59 | 7 | 93 | 0 | 0 | 178 | 5.5 |
| 34 | 01/10/17 | 1606-1617 | 34.59, 35.23 | 01/10/17 | 0958 (N) | 33.81, 35.18 | 14 | 0 | 86 | 0 | 74 | 7 |
| 35 | 06/10/17 | 2002-2014 | 16.35, 16.69 | 07/10/17 | 0031 (D) | 16.22, 16.65 | 24 | 61 | 15 | 0 | 60 | 3 |
| 36 | 06/10/17 | 2157-2218 | 10.7, 12.28 | 06/10/17 | 1341 (N) | 10.12, 12.36 | 0 | 100 | 0 | 0 | 141 | 11 |
| 37 | 07/10/17 | 0121-0132 | -4.44, -3.48 | 06/10/17 | 1345 (N) | -3.46, -2.52 | 0 | 100 | 0 | 0 | 230 | 1 |
| 38 | 14/10/17 | 1246-1311 | -60.1, -58.69 | 14/10/17 | 1824 (D) | -58.97, -57.04 | 14 | 86 | 0 | 0 | 302 | 2.5 |
| 39 | 17/10/17 | 1547-1607 | -25.72, -24.23 | 17/10/17 | 1546 (D) | -26.95, -24.01 | 22 | 78 | 0 | 0 | 112 | 0 |
| 40 | 20/10/17 | 1428-1448 | 28.04, 29.79 | 20/10/17 | 0355 (N) | 26.51, 28.96 | 0 | 100 | 0 | 0 | 210 | 16 |
| 41 | 20/10/17 | 1527-1546 | 34.3, 36 | 20/10/17 | 0353 (N) | 33.84, 36.77 | 0 | 100 | 0 | 0 | 140 | 14.5 |
| 42 | 23/10/17 | 1847-1859 | 44.8, 45.32 | 24/10/17 | 0642 (N) | 44.03, 48.97 | 5 | 58 | 36 | 0 | 290 | 5 |
| 43 | 24/04/18 | 1905-1929 | 3.53, 4.97 | 24/04/18 | 2200 (D) | 2.04, 5.98 | 15 | 75 | 10 | 0 | 425 | 9 |
| 44 | 29/04/18 | 2124-2152 | 42.48, 44.8 | 29/04/18 | 1302 (N) | 41.01, 44.95 | 13 | 32 | 27 | 28 | 231 | 16.5 |
| 45 | 01/05/18 | 2119-2138 | 10.71, 12.14 | 01/05/18 | 1258 (N) | 10.01, 12.96 | 0 | 85 | 0 | 15 | 188 | 0.5 |
| 46 | 01/05/18 | 2221-2245 | 4.39, 6.19 | 01/05/18 | 1300 (N) | 3.04, 6.98 | 0 | 88 | 0 | 12 | 385 | 6 |
| 47 | 01/05/18 | 2328-0011 | -2.02,-0.19 | 02/05/18 | 0123 (D) | -1.96, -0.22 | 0 | 100 | 0 | 0 | 71 | 2 |
| 48 | 06/05/18 | 2021-2033 | -43.59, -43.42 | 06/05/18 | 1332 (N) | -45.96, -43.01 | 12 | 88 | 0 | 0 | 202 | 9 |
| 49 | 07/05/18 | 0254-0318 | -64.72, -64.27 | 07/05/18 | 0746 (N) | -65.18, -63.02 | 0 | 100 | 0 | 0 | 99 | 2 |
| 50 | 12/05/18 | 1536-1603 | -37.22, -35.4 | 12/05/18 | 1640 (D) | -37.97, -35.41 | 29 | 71 | 0 | 0 | 177 | 0 |
| 51 | 12/05/18 | 1932-1956 | -16.84, -15 | 13/05/18 | 0337 (N) | -16.98, -14.52 | 28 | 72 | 0 | 0 | 107 | 11 |
| 52 | 14/05/18 | 1649-1712 | 20.87, 22.63 | 14/05/18 | 1505 (D) | 22.02, 23.99 | 5 | 95 | 0 | 0 | 325 | 10 |
| 53 | 14/05/18 | 1932-1952 | 38.06, 38.95 | 14/05/18 | 1510 (D) | 38.52, 39.49 | 52 | 26 | 22 | 0 | 95 | 14.5 |