# Peer review of "Evaluation of aerosol number concentrations from CALIPSO with ATom airborne in-situ measurements"

_Atmospheric Chemistry and Physics, 2022_

## Author Comment (AC1)

We thank the reviewers for their time and effort in reviewing our manuscript. We found their comments to be very helpful in enhancing the quality of our article. Our point-by-point replies to the comments are provided below. Referee comments are given in black, our answers are given in blue.

**Reviewer-1:**

1) Lines 15-18. Correlation coefficients are used to evaluate "good agreement". Correlation is correlation, not quantitative agreement, which is better estimated by RMSE and bias.

Thank you for pointing this out. We have now also mentioned the RMSE and bias values along with the correlation coefficient in the revised manuscript as follows.

"The POLIPHON estimates of $n_{50,dry}$ and $n_{250,dry}$ were found to be in good agreement with the in-situ measurements with a correlation coefficient of 0.829 and 0.47, root mean square error (RMSE) of 234 cm$^{-3}$ and 13 cm$^{-3}$, bias of -97 cm$^{-3}$ and 4 cm$^{-3}$, respectively. The OMCAM estimates of $n_{50,dry}$ and $n_{250,dry}$ were also in reasonable agreement with the in-situ measurements with a correlation coefficient of 0.823 and 0.463, RMSE of 247 cm$^{-3}$ and 13 cm$^{-3}$, bias of 44 cm$^{-3}$ and 4 cm$^{-3}$, respectively."

2) Section 2.3.2 (OMCAM): The relationship given by Eq. 3 must depend very sensitively on the choices for the size distribution modes described in Eq. 2, especially for sizes in the CCN-active region (r>0.025 µm) where the size distribution is often very steeply sloped. It would be useful for this manuscript to provide more information, in the form of a table, on the choices for the lognormal parameters for the different aerosol types that were used. Some discussion on their variability would also be useful.

Thank you for your suggestion. As pointed out in the manuscript, the lognormal size distribution parameters and the overall OMCAM retrieval algorithms are discussed comprehensively in Choudhury and Tesche (2022). We agree that including the size distribution parameters in the current manuscript will improve its completeness. Thus, we now include them in Table 3. In this paper, the authors also performed sensitivity tests by varying the initial normalized size distributions to account for the natural variability and found the uncertainty to be a factor of 2. We have added the following text to the manuscript.

"Since the algorithm primarily relies on the assumption of fixed initial normalized size distributions for every aerosol subtype, Choudhury and Tesche (2022) analyzed the sensitivity of the output $n_{j,dry}$ to variations of these size distributions. By varying the magnitude of the fine and coarse modes of the size distributions by ±50%, they found the resulting $n_{j,dry}$ to remain within a factor of 2. Such an uncertainty is expected for a spaceborne retrieval of aerosol microphysical properties and is also similar to that of POLIPHON (Mamouri and Ansmann, 2016)."

3) Lines 171-173: I'm not sure what is meant by "using the microphysical properties by Sayer et al. (2012) in the OMCAM algorithm and estimate the ANC separately". The OMCAM algorithm is used to derive the ANC via Eq. 3; how can you estimate the ANC "separately"? Perhaps it's better to explain that a different marine model is applied in Sect. 3.2.1 upon finding that the Sayer et al. model produced significant biases when compared with the in situ data. At least that's clear, if perhaps providing a bit too much foreshadowing of results.

We apologize for the confusion. The OMCAM algorithm as per Choudhury and Tesche (2022) uses the aerosol size distributions and refractive indices from the CALIPSO aerosol

model (Omar et al., 2009). As we found the $n_{50,dry}$ estimated using the marine model from Omar et al. (2009) to be inconsistent with the in-situ measurements, we used the recent AERONET based marine model by Sayer et al. (2012) in the OMCAM algorithm and re-computed the aerosol number concentrations which were then found to be in better agreement. We have rephrased the sentence and added some additional information:

"In this work, we also utilize the microphysical properties for marine aerosols as recommended by Sayer et al. (2012) in the OMCAM algorithm to examine its potential in deriving the ANC from CALIOP measurements (presented in Section 3.2.1). The size distribution parameters along with the complex refractive indices are listed in Table 3. Please note that the parameters in Sayer et al. (2012) are given for ambient conditions and were converted to dry conditions assuming a uniform RH of 70 % before using them in the OMCAM algorithm. The size distribution was modified to dry conditions by using kappa parametrization (Petters and Kreidenweis, 2007) and the refractive index was modified as per the volume weighting rule. "

4) Summary: One of the take-home messages for me was that, while these algorithms do surprisingly well for the r>50 aerosol number concentration, they are pretty sensitive to assumptions about the size distribution model in the training set. Clearly there needs to be a better understanding of the variability in space and time of these parameters. So comparison with measurements such as Schmale et al. (2017) would be useful, but I would think a more comprehensive evaluation of the range of size distribution parameters in different airmass types would be an important long-term objective. There are many potential datasets out there; for example, for the marine aerosol, there is an extensive ship-borne global dataset from Quinn et al. (2017), as well as an airborne datasets by Clarke and Kapustin (2002) and Clarke et al. (2010). A more comprehensive survey may be in order to thoroughly bound uncertainties in the space-borne retrieval of ANC and CCN. Certainly beyond the scope of this work, but perhaps a target for the future.

Thank you for your suggestions. We have included and discussed all the suggested articles in the updated manuscript. The following sentences are added to the summary:

"Future work includes a direct comparison of the type-specific extinction-to-number concentration conversion factors from in-situ measurements, for instance from Brock et al. (2021), with POLIPHON. The aerosol size distributions used in OMCAM can also be compared with in-situ measurements (e.g. Clarke and Kapustin, 2002, 2010; Quinn et al., 2017; Brock et al., 2021) so as to better quantify the uncertainty associated with the output aerosol number concentrations for different aerosol subtypes."

**Reviewer-2:**

1. Fig.3b. "Age of parcel". Units are not shown

Thank you for pointing it out. We have now included it in the revised manuscript.

2. Fig.4(a-d). From where the extinction profiles of dust, smoke, etc are taken from? Is it CALIPSO product? Are these extinctions for dry particles? When separating the aerosol types, was the dependence of depolarization on RH considered?

We apologize for the confusion. We use the aerosol subtype mask included in the CALIPSO data product (discussed in Section 2.2) to separate the altitude bins for different dominant

(highest extinction coefficient) aerosol subtypes. However, the situation is different for mixed aerosol subtypes (dusty marine and polluted dust). As stated in the lines 120-122, we separate the aerosol mixtures in the very first step of our algorithm as per the methodology given in Tesche et al. (2009) using the depolarization ratio and lidar ratios. We haven't considered the dependence of depolarization ratio on the RH. We have added the following in the modified manuscript.

"However, it does not consider the RH dependency of depolarization ratio which may result in additional errors especially in marine environments."

**Additional modifications:**

1. Table 3 is now changed to Table 4.

2. The conversion factors for the modified marine model in Table 4 are now corrected. The previous conversion factors were mistakenly computed for ambient conditions. However, this doesn't affect other results as these conversion factors were specifically computed to simplify the use of OMCAM algorithm for future works. The authors use look-up tables of pre-computed parameters to estimate $n_{j,\mathrm{dry}}$ to avoid truncation errors.